# Uni[MASK]:
# Unified Inference in Sequential Decision Problems

**Micah Carroll**[1], **Orr Paradise**[1], **Jessy Lin**[1], **Raluca Georgescu**[2], **Mingfei Sun**[2], **David Bignell**[2], **Stephanie Milani**[3], **Katja Hofmann**[2], **Matthew Hausknecht**[2], **Anca Dragan**[1], and **Sam Devlin**[2]

[1]UC Berkeley
[2]Microsoft Research
[3]CMU

## Abstract

Randomly masking and predicting word tokens has been a successful approach in pre-training language models for a variety of downstream tasks. In this work, we observe that the same idea also applies naturally to sequential decision making, where many well-studied tasks like behavior cloning, offline reinforcement learning, inverse dynamics, and waypoint conditioning correspond to different sequence maskings over a sequence of states, actions, and returns. We introduce the Uni[MASK] framework, which provides a unified way to specify models which can be trained on many different sequential decision making tasks. We show that *a single* Uni[MASK] *model* is often capable of carrying out many tasks with performance similar to or better than single-task models. Additionally, after fine-tuning, our Uni[MASK] models consistently outperform comparable single-task models. Our code is publicly available here.

## 1 Introduction

Masked language modeling [11] is a key technique in natural language processing (NLP). Under this paradigm, models are trained to predict randomly-masked subsets of tokens in a sequence. For example, during training, a BERT model might be asked to predict the missing words in the sentence "yesterday I ___ cooking a ___". Importantly, while unidirectional models like GPT [33] are trained to predict the next token conditioned only on the left context, bidirectional models trained on this objective learn to model both the left *and* right context to represent each word token. This leads to richer representations that can then be fine-tuned to excel on a variety of downstream tasks [11].

Our work investigates how masked modeling can be a powerful idea in sequential decision problems. Consider a sequence of states $s$ and actions $a$ collected across $T$ timesteps $s_1, a_1, \ldots, s_T, a_T$. If we consider each state and action as tokens of a sequence (analogous to words in NLP) and mask the last action $(s_1, a_1, s_2, a_2, s_3, \_)$, then predicting the missing token $a_3$ amounts to a Behavior Cloning prediction with two timesteps of history [32], given that this masking corresponds to the inference $\mathbb{P}(a_3|s_{1:3}, a_{1:2})$. From this perspective, training a model to predict missing tokens from all maskings of the form $(s_1, a_1, \ldots, s_t, \_, \ldots, \_)$ for all $t \in [1, \ldots, T]$ corresponds to training a Behavior Cloning (BC) model.

In this work, we introduce the Uni[MASK] framework: **Uni**fied Inferences in Sequential Decision Problems via [MASK]ings, where inference tasks are expressed as *masking schemes*. In this framework, commonly-studied tasks such as goal or waypoint conditioned BC [12, 36], offline reinforcement learning (RL) [25], forward or inverse dynamics prediction [18, 9, 6], initial-state inference [38], and others are unified under a simple sequence modeling paradigm. In contrast to standard approaches that train a model for each inference task, we show how this framework naturally lends itself to multi-task

36th Conference on Neural Information Processing Systems (NeurIPS 2022).

training: a single Uni[MASK] model can be trained to perform a variety of tasks out-of-the-box by appropriately selecting sequence maskings at training time.

We test this framework in a Gridworld navigation task and a continuous control environment. First, we train a Uni[MASK] model by sampling from the space of all possible maskings at training time (random masking) and show how this scheme enables a single Uni[MASK] model to perform BC, reward-conditioning, waypoint-conditioning, and more by conditioning on the appropriate subsets of states, actions, and rewards. We then systematically analyze how the masking schemes seen at training time affect downstream task performance. Training on random masking generally does not compromise single-task performance, and in fact can outperform models that only train on the task of interest. In the continuous control environment, we confirm that a model trained with random masking and fine-tuned on BC or RL tends to outperform models specialized to those tasks.

Our results suggest that expressing tasks as sequence maskings with the Uni[MASK] framework may be a promising unifying approach to building general-purpose models capable of performing many inference tasks in an environment [2], or simply offer an avenue for building better-performing single-task models via unified multi-task training.

In summary, our contributions are:

1. We propose a new framework, Uni[MASK], that unifies inference tasks in sequential decision problems as different masking schemes in a sequence modeling paradigm.
2. We demonstrate how randomly sampling masking schemes at training time produces a single multi-inference-task model that can do BC, reward-conditioning, dynamics modeling, and more out-of-the-box.
3. We test how training on many tasks affects single-task performance and show how fine-tuning models trained with random masking consistently outperforms single-task models.
4. We show how the insights we have gained while developing our choice of Uni[MASK] architecture can be used to improve other state-of-the-art methods.

## 2 Related Work

**Transformer models.** The great successes of transformer models [41] in other domains such as NLP [11, 33, 3] and computer vision [13, 19] motivates our work. Using transformers in RL and sequential decision problems has proven difficult due to the instability of training [30], but recent work has investigated using transformers in model-based RL [6], motion forecasting [29], learning from demonstrations [34], and teleoperation [10]. We focus on developing a unifying framework interpreting tasks in sequential decision problems as maskings.

**The utility of masked prediction.** Work in both NLP [11] and vision [5, 19] have explored how masked prediction is useful as a self-supervision task. In the context of language generation, [39] provides a framework for thinking about different masking schemes. Recent work has also explored how random masking can be used to do posterior inference in a probabilistic program [43].

**Sequential decision-making as sequence modeling.** Previous and concurrent work [7, 21, 24] shows how to use GPT-style (causally-masked) transformers to directly generate high-reward trajectories in an offline RL setting. We expand our focus to many tasks that a sequence modeling perspective enables, including but not restricted to offline RL. Although previous work has cast doubt on the necessity of using transformers to achieve good results in offline RL [14], we note that offline RL [25] is just one of the various tasks we consider. Concurrent work generalizes the left-to-right masking in the transformer to condition on future trajectory information for tasks like state marginal matching [17] and multi-agent motion forecasting [29]. In contrast, we systematically investigate how a single bidirectional transformer can be trained to perform arbitrary downstream tasks in more complex settings than motion forecasting – i.e., we also consider agent actions and rewards in addition to states. The main thing that sets us apart from these works is a systematic view of all tasks that can be represented by this sequence-modeling perspective, and a detailed investigation of how different multi-task training regimes compare.

**Prior work on tasks in sequential decision problems.** While we use masked prediction as the self-supervision objective, previous work on self-supervised learning for RL has investigated other auxiliary objectives, such as state dynamics prediction [37] or intrinsic motivation [31]. Typically,

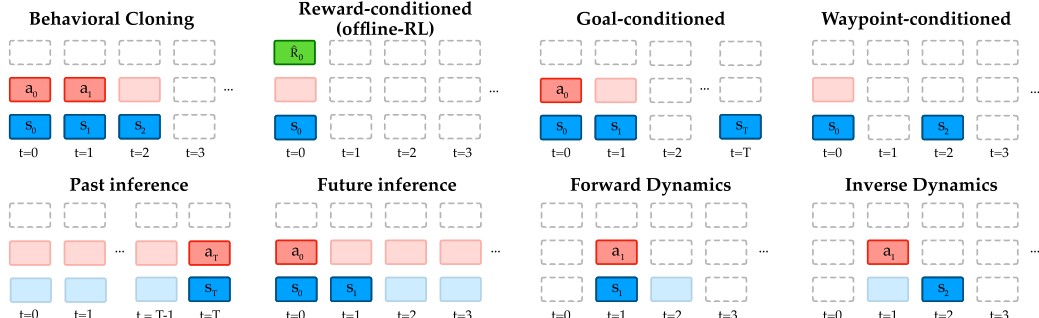

Figure 1: Uni[MASK] **framework: Representing arbitrary tasks as masking schemes.** For each task, we show the inputs to the model (solid colors) and the outputs the model must predict (translucent colors). For example, in future inference, the model must predict all future states and actions conditioned on the initial states and actions. Here we only display one input masking scheme for each task, but many tasks are fully represented by multiple masking schemes. For example, BC has up to T different masking schemes, one for each possible history length (although in practice one would generally use the model with a sliding window).

to accomplish the tasks we consider, prior work relies on single-task models: for example, goal-conditioned imitation learning [12], RL [22], waypoint-conditioning [36], property-conditioning [45, 17], or dynamics model learning [18, 9]. Other work has focused on training models to perform different "tasks" such as different games in Atari [24] or different environments and multi-modal prediction tasks [35]. In contrast, we are interested in performing different *inference* tasks in a single environment, such as RL and forward dynamics modeling, using sequence modeling as a unifying framework.

## 3  The Uni[MASK] Framework

We introduce the Uni[MASK] framework. In Section 3.1 we propose a unifying interpretation of inference tasks in sequential decision problems as masking schemes. In Section 3.2 we describe different ways of training Uni[MASK] models, and provide hypotheses about their efficacy.

We consider trajectories as sequences of states, actions, and optionally property tokens (e.g. reward): $\tau = \{(s_0, a_0, p_0), \ldots, (s_T, a_T, p_T)\}$.[1]

Motivated by canonical problems in decision-making that involve reward, in most of our analysis we use return-to-go (RTG) as the property (the sum of rewards from timestep $t$ to the end of the episode): that is, we set $p_t = \hat{R}_t$ where $\hat{R}_t = \sum_{t'=t}^{T} r_{t'}$. However, any property of the decision problem can be considered a "property token," including specific environment conditions being satisfied, the style of the agent, or the performance of the agent (e.g. the reward obtained in the timestep). In order to train on tasks requiring specific properties, one must have labels for them – obtained either programmatically or through human annotators. We demonstrate how our model can be conditioned on a non-reward property in Appendix F.

### 3.1  Tasks as Masking Schemes

In the Uni[MASK] framework, we formulate tasks in sequential decision problems as input masking schemes. Formally, a masking scheme specifies which input tokens are masked (determining what tokens are shown to the model for prediction) and which outputs of the model are masked before computing losses (determining which outputs the model should learn to predict). For example, the masking scheme for BC unmasks (conditions on) $s_{0:t}$ and $a_{0:t-1}$, and the model must predict $a_t$.

In Figure 1, we illustrate how to unify commonly-studied tasks such as BC, goal and waypoint conditioned imitation, offline RL (reward-conditioned imitation), and dynamics modeling under our proposed representation of tasks as masking schemes. We describe the masking scheme for each of these tasks in detail in Appendix B.

---

[1]While reward-to-go (or other trajectory statistics) are not necessary, we formulate the most general form to showcase how one can easily condition on additional available properties of a trajectory. Using reward-to-go also enables us to compare our method with previous offline-RL work [7].

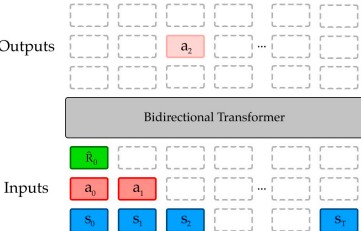

Figure 2: The Uni[MASK] model takes in a snippet of a trajectory which is masked according to a masking scheme before inference time. For each input possible masking, there are (many) corresponding tasks of predicting the missing inputs. Above we show an input masking corresponding to conditioning on both reward-to-go *and* final (goal) state; we highlight the output corresponding to predicting the agent's next action, i.e. performing the inference $\mathbb{P}(a_2 \mid s_{0:2,T}, a_{0:1}, \hat{R}_0)$.

## 3.2 Model Architecture & Training Regimes

For our main experiments, we instantiate our Uni[MASK] framework using the BERT architecture [11] adapted to the sequential decision problem domain, consisting of a positional encoding layer and stacked bidirectional transformer encoder (self-attention) layers (see Figure 2). One key difference with the original BERT architecture is that we stack the state, action, and property (e.g. RTG) tokens for each timestep into a single vector. While prior work in sequential decision-making had used timestep encoding [7] (which can be thought of as concatenating each observation with its environment timestep), we found traditional positional encoding [41, 11] to reduce overfitting. For reward-conditioned tasks, in each context window we only feed the first RTG token into the model along with the number of timesteps remaining in the horizon. This information is sufficient for reward-conditioning at inference time, and we found that it outperformed the standard approach of feeding in the RTG token at every timestep [7, 21]. See Appendix D for more model details, and Appendix F for experiments with an alternative instantiation of Uni[MASK] with a feedforward neural network architecture.

### 3.2.1 Training regimes

We experiment with four ways to train a Uni[MASK] model on masked prediction, illustrated in Figure 3 and described below.

`single-task`. Training on just one of the masking scheme described in Section 3.1.

`multi-task`. Training a single model on multiple masking schemes: each trajectory snippet is masked according to one of the schemes from Section 3.1 (chosen at random).

> *Intuition: Could allow a single model to perform well on multiple tasks. Additionally, it might outperform `single-task` on individual tasks, as the model could learn richer representations of the environment from the additional masking schemes.*

`random-mask`. Training a single model on a fully randomized masking scheme. For each trajectory snippet, first, a masking probability $p_{\text{mask}} \in [0, 1]$ is sampled uniformly at random; then each state and action token is masked with probability $p_{\text{mask}}$; lastly, the first RTG token is masked with probability $1/2$ and subsequent RTG tokens are always masked (see Appendix C for details).

> *Intuition: Could allow a single model to perform well on **any** sequence inference task without the need to specify the tasks of interest at training time. The model may learn richer representations than those of `multi-task` as it must reason about **all** aspects of the environment.*

`finetune`. Fine-tune a model pre-trained in `random-mask` on a specific masking scheme.

> *Intuition: Performing fine-tuning could allow the model to benefit from the improved representations obtained from `random-mask`, while specializing to the single task at hand.*

### 3.2.2 Hypotheses

Based on the intuition of the strengths of each training regime, we formulate the following hypotheses:

**H1.** First training on multiple inference tasks will lead to better performance on individual tasks than only training on that inference task: {`multi-task`, `random-mask`, `finetune`} > `single-task`.

**H2.** Randomized mask training outperforms training on a specific set of tasks: `random-mask` > `multi-task`.

**H1** tests whether models learn richer representations by training on multiple inference tasks. **H2** tests a stronger claim: whether training on *all* possible tasks by randomly sampling masking at training time is better than selecting a set of specific maskings.

# 4 A Unified Model for Any Inference Task

We first demonstrate how `random-mask` enables a single Uni[MASK] model to perform arbitrary inference tasks at test-time on a Gridworld environment, without the need for task-specific output heads or training schemes that are customized for the downstream task. We then show that `random-mask` does not compromise performance on most specific tasks of interest. Models trained with `random-mask` achieve comparable or better performance to `single-task` and `multi-task`-models, and in fact consistently outperform after additional fine-tuning on the task of interest (`finetune`).

**Environment Setup.** We design a fully observable $4 \times 4$ Gridworld in which the agent should move to a fixed goal location behind a locked door with the MiniGrid environment framework [8]. The agent and key positions are randomized in each episode. The agent receives a reward of $1$ for each timestep it moves closer to the goal, $-1$ if it moves away from the goal, and $0$ otherwise. We train Uni[MASK] models on training trajectories of sequence length $T = 10$ from a noisy-rational agent [46]. More detailed information about the environment is in Appendix E.

## 4.1 One Model to Rule Them All

As shown in Figure 4, a single Uni[MASK] model trained with `random-mask` can be used for arbitrary inference tasks by conditioning on specific sets of tokens. Unless otherwise indicated, we take the highest probability action from the model $a_t = \arg\max_{a'_t} \mathbb{P}(a'_t \mid s_0, a_0, \dots, s_t)$, and then query the environment dynamics for the next state $s_{t+1}$. The model can be used for imitation, reward- and goal-conditioning, or as a forward or inverse dynamics model (when querying for state predictions, as in the backwards inference task). If trajectories are labeled with properties at training time, the model can also be used for property-conditioning. In Figure 5, we show how the model can also be conditioned on *global* properties of the trajectory, such as whether the trajectory passes through a certain position at any timestep.

Qualitatively, these results suggest that the model generalizes across masking schemes, since seeing the exact masking corresponding to a particular task at training time is exceedingly rare (out of $2^T \times 2^T \times 2$ possible state, action, and RTG maskings for a sequence of length $T$).

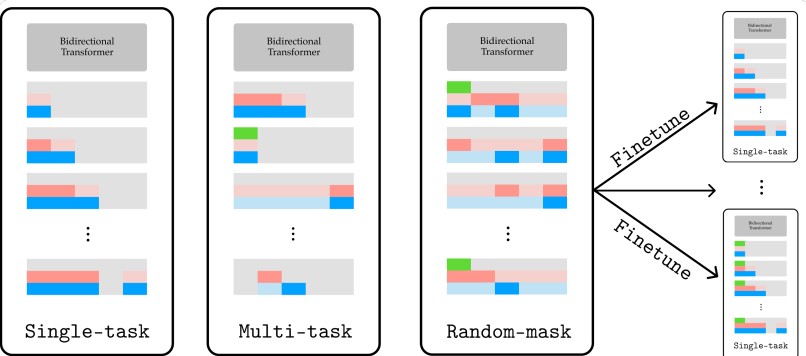

Figure 3: **The four training regimes considered in this work:** `single-task`, `multi-task`, `random-mask` and `finetune`.

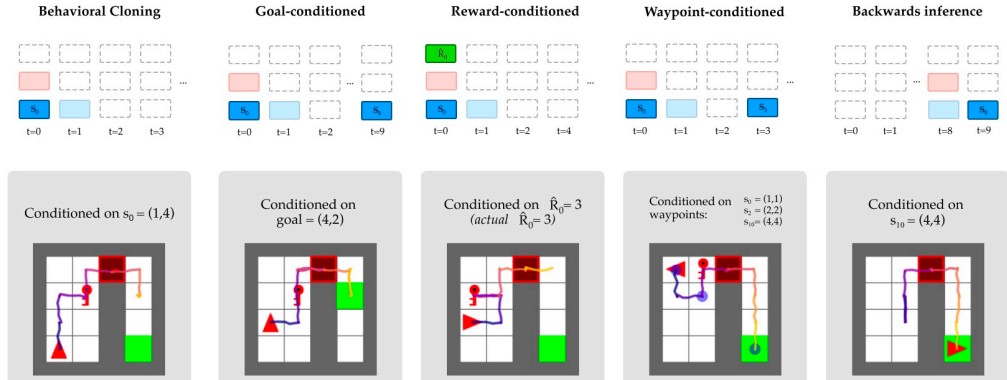

Figure 4: **A** Uni[`MASK`] **model trained with random masking queried on various inference tasks. (1) Behavioral cloning:** generating an expert-like trajectory given an initial state. **(2) Goal-conditioned:** reaching an alternative goal. **(3) Reward-conditioned:** generating a trajectory that achieves a particular reward. **(4) Waypoint-conditioned:** reaching specified waypoints (or subgoals) at particular timesteps, e.g. going down on the first timestep instead of immediately picking up the key. **(5) Backwards inference:** generating a likely *history* conditioned in a final state (by sampling actions and states backwards). Trajectories are shown with jitter for visual clarity.

## 4.2 Future State Predictions

Uses of the `random-mask`-trained Uni[`MASK`] model are not limited to rolling out new trajectories (requesting inferences about the agent's next action). One can also request inferences for states and actions further into the future: e.g., "where will the agent be *in 3 timesteps*?". Given a fixed initial set of observed states, we visualize the distribution of predicted states at each timestep in Figure 6. Since we do not roll out actions, querying the model for the predicted state distribution at a particular timestep marginalizes over missing actions; for example, $\mathbb{P}(s_1 \mid s_0, s_3, s_6)$ models the possibility that the agent chooses either up or left as the first action. Qualitatively, the state predictions suggest that the model accurately captures the environment dynamics and usual agent behavior; e.g. it correctly models that the agent has equal probability of going up and right at $t = 3$ (leading it to the distribution over states at $t = 4$), and that the agent must be at position $(2, 1)$ at $t = 5$ to reach the door at $t = 6$.

## 4.3 Measuring Single-Task Performance

Next, we investigate how a `random-mask` model performs on individual tasks, in comparison to `single-task` models trained exclusively on the evaluated task. If we care about a single task (e.g. goal-conditioned imitation), should we train a model simply on that task? Or can there be advantages to training a general model first, and then fine-tuning it to the task of interest? For this set of experiments, we primarily consider validation loss as our measure of performance. Validation loss

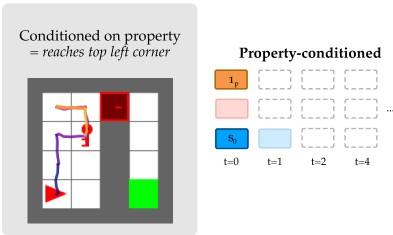

Figure 5: **Other types of property-conditioning.** If the training dataset has additional property labels (e.g., whether the trajectory passes by the top left corner of the grid *at any timestep*), the model can roll out trajectories conditioned on whether the property is exhibited.

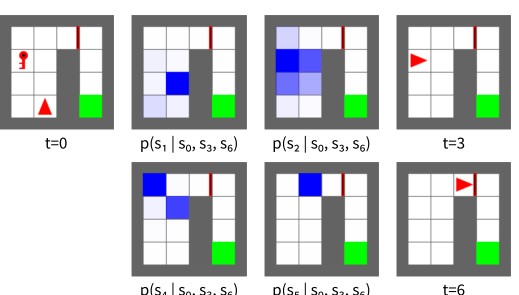

Figure 6: **Predicted state distributions.** The model is conditioned on states at $t = 0, 3, 6$.

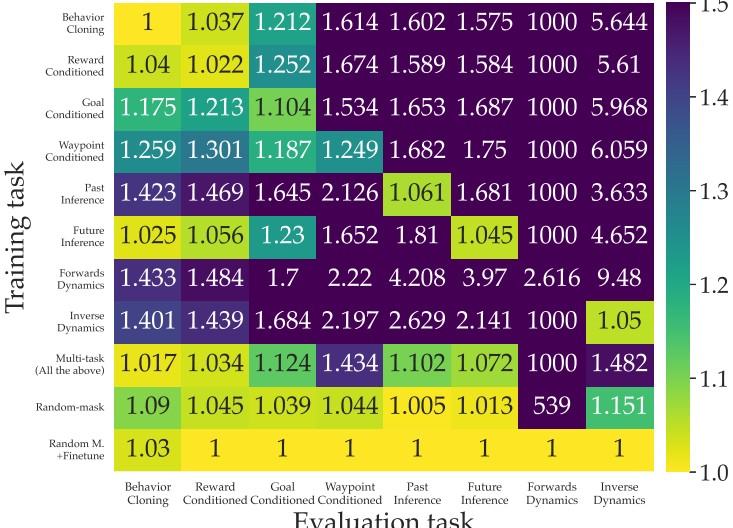

Figure 7: **Task-specific validation losses (normalized column-wise).** Each row corresponds to the performance of a single model evaluated in various ways, except for the last row—for which each cell is fine-tuned on the respective evaluation task. Loss values are averaged across six seeds and then divided by the smallest value in each column. Thus, for each evaluation task (i.e., column), the best method has value 1; a value of 1.5 corresponds to a loss that is 50% higher than the best model in the column. Note that the performance of a multi-task model on the forward dynamics task is particularly poor since the environment is deterministic: we should expect overfitting (with a single-task model) to perform the best. See Appendix F for more details.

provides a general way to evaluate how well models fit the distribution of trajectories and transitions, which is what we are concerned with for most inference tasks: i.e., "how well can the network predict the true state or action in the data?".

In Figure 7, we report validation loss if the model is trained on one task (or multiple tasks) and evaluated on another task. As expected, models trained on one masking (e.g. BC) perform well when queried on the task they were trained on (as seen on the diagonal), but poorly when queried with another task (e.g. past inference).

First, we find that `random-mask` training (but not `multi-task` training) outperforms `single-task` training on half of the tasks considered, showing that even if one is interested in a single inference task, training on many more tasks can sometimes improve performance. Specializing a model trained with `random-mask` via finetuning (`finetune`) leads to the best performance, outperforming `single-task` on all tasks except behavior cloning. This means that even if one is interested in a single inference task, first training on multiple tasks generally improves performance. Overall, these results do not fully support **H1**, given `multi-task`'s poor performance and `random-mask`'s performance which is not consistently better than `single-task`.

We also find that `random-mask` training leads to lower loss values on almost all evaluation tasks relative to `multi-task`, supporting **H2**: training on additional inference tasks beyond the specific ones of interest can augment performance.

# 5 Trajectory Generation in a Complex Environment

In addition to Gridworld, we test our method in a partially observable, continuous-state and continuous-action environment, with a larger trajectory horizon (200 timesteps).

## 5.1 Environment Setup

We adapt the Mujoco-physics Maze2D environment [16] (see Appendix H for figures), in which a point-mass object is placed at a random location in a maze, and the agent is rewarded for moving

towards a randomly generated target location (making this task "goal-conditioned by default"). We make this task harder by removing the agent's velocity information from each timestep's observation and increasing the amount of initial position randomization. These changes make the environment partially observable, forcing models trained on this data to implicitly infer the agent's velocity from observed context.

**Expert dataset.** We want our expert data to have some suboptimality so that reward-conditioning can be tested for better-than-demonstrator performance. We generate a dataset of expert trajectories by rolling out D4RL's PD controller (which is non-Markovian), and add noise to the actions with zero-mean and $0.5$ variance (which are then clipped to have each dimension between $-1, 1$). We generate 1000 trajectories of 200 timesteps, of which 900 are used for testing and 100 for validation. For more details on our adapted Maze2D environment and design decisions, see Appendix H.

## 5.2 Models Trained

For the Maze2D evaluations, we focus on test-time reward performance on behavior cloning and offline RL (reward-conditioning) across various architectures and training regimes.

We consider Uni[MASK] models trained with the different training regimes: `single-task`, `multi-task`, `random-mask`, and `finetune`. We additionally consider other architectures, such as a feed-forward NN and Decision Transformer (DT) baselines [7]. We found that several of our design decisions for Uni[MASK] models – using positional encoding instead of timestep encoding, inputting the return-to-go token at the first timestep with the number of timesteps in the horizon – also improved GPT-based models like DT. We call our improved baseline Decision-GPT (for implementation details, see Appendix G). We train our Decision-GPT model with the `single-task` training regime. The only meaningful difference between Decision-GPT and a `single-task` Uni[MASK] model is whether the model is GPT- or BERT-based.

For each architecture and applicable training regime, we train separate models to perform behavior cloning and offline RL (reward-conditioning). The only exceptions are Uni[MASK] models trained with `multi-task` (trained to perform BC and RC) and `random-mask`. We train two sets of such models, for context lengths of 5 and 10 – meaning that during both training and evaluation, the models will respectively only be able to see the last 5 or 10 timesteps of the agent's interaction with the environment.

## 5.3 Results

We report reward evaluation results for 1000 rollouts in the Maze environment with standard errors across 5 seeds in Table 1.

**The value of pre-training and fine-tuning for** Uni[MASK] **models.** We find that fine-tuning is critical for good performance in more complex environments. `multi-task` performs more-or-less comparably to `single-task` in behavior cloning and reward conditioning; however, `random-mask` in this setting obtains significantly lower rewards (counter to **H2**). This suggests that `multi-task` training can be effective in mostly maintaining reward performance while increasing the breadth of functionality, but training on too many tasks can hurt out-of-the-box performance. However, `finetune` recovers the performance loss, again out-performing `single-task` (providing qualified support for **H1**). Surprisingly, for a context length of ten, fine-tuning `multi-task` does not improve performance as much as fine-tuning the randomly masked model, suggesting that specifically training on random masking might provide benefits for adapting models to individual downstream tasks.

**How do** Uni[MASK] **models compare to other architectures?** For context length five, we see that multi-task with finetuning and `finetune` Uni[MASK] models perform better than all baselines we consider. However, increasing the context length to ten, we see that Uni[MASK] models performs poorly across the board, with the finetuned conditions outperformed by our Decision-GPT baseline. We speculate that this might be related to the documented difficulty of using BERT-like architectures (as that of Uni[MASK] models) for sequence generation [42, 28].

**Isolating the effect of GPT vs. BERT.** In order to investigate the effect of using GPT-like architectures instead of BERT-like ones, we can consider the comparison between `single-task` Uni[MASK] and our Decision-GPT baseline: the main difference between these two models is only whether one

uses BERT or GPT as the backbone of the architecture.[2] We find that while using GPT seems to yield similar (or worse) performance to BERT for context length five, using GPT seems to give an advantage for longer sequence lengths. In particular, note that a larger context length enables GPT to increase performance, while performance worsens for `single-task` Uni[`MASK`]. This suggests that if one were able to use a GPT architecture and train it with random masking and fine-tuning, it might be possible to get the best of both worlds.

Table 1: **Maze2D Results. Comparing among** Uni[`MASK`] **models**, we isolate the benefit of `finetune`: this training regime tends to perform best across tasks and sequence lengths. **Comparing** `single-task` Uni[`MASK`] **to our Decision-GPT model**, we can isolate the effect of using a BERT-like architecture vs. a GPT-like architecture: for larger context lengths, BERT-like models struggle to maintain the same generation quality. Every entry in the table corresponds to a separate model, except for the cells denoted with $^\dagger$, which use the same model across tasks (but not sequence lengths).

| Model | Context Length 5 | | Context Length 10 | |
|---|---|---|---|---|
| | BC | RC | BC | RC |
| Uni[`MASK`] **Models** | | | | |
| Uni[`MASK`]-`single-task` | $2.66 \pm 0.03$ | $2.64 \pm 0.02$ | $2.47 \pm 0.04$ | $2.41 \pm 0.05$ |
| Uni[`MASK`]-`multi-task` (BC & RC) | $2.65 \pm 0.01^\dagger$ | $2.68 \pm 0.01^\dagger$ | $2.39 \pm 0.03^\dagger$ | $2.39 \pm 0.03^\dagger$ |
| Uni[`MASK`]-`multi-task` + finetune | $2.73 \pm 0.01$ | $2.74 \pm 0.01$ | $2.42 \pm 0.04$ | $2.42 \pm 0.03$ |
| Uni[`MASK`]-`random-mask` | $2.19 \pm 0.09^\dagger$ | $2.20 \pm 0.09^\dagger$ | $2.29 \pm 0.07^\dagger$ | $2.31 \pm 0.06^\dagger$ |
| Uni[`MASK`]-`finetune` | $2.67 \pm 0.03$ | $2.73 \pm 0.01$ | $2.55 \pm 0.03$ | $2.61 \pm 0.03$ |
| **Other architectures** | | | | |
| Feedforward Neural Network | $1.68 \pm 0.07$ | $1.53 \pm 0.08$ | $1.83 \pm 0.06$ | $1.88 \pm 0.06$ |
| Decision Transformer [7] | $1.13 \pm 0.07$ | $1.49 \pm 0.04$ | $1.58 \pm 0.06$ | $1.70 \pm 0.07$ |
| Our Decision-GPT model | $2.66 \pm 0.01$ | $2.32 \pm 0.05$ | $2.74 \pm 0.01$ | $2.73 \pm 0.02$ |

## 6   Limitations and Future Work

**Comparison to other specialized models.** We show that Uni[`MASK`] outperforms feedforward networks, Decision Transformer models, and for short sequence lengths also our own improved GPT-based baseline. However, we do not compare our models directly to different models in prior work that are specialized for specific tasks (e.g. goal-conditioning models, etc.). While this is a limitation of our work, it is also not our main focus: we propose a unifying framework for a variety of tasks in sequential decision problems, and extensively analyze how different training regimes affect performance.

**Longer context lengths.** One limitation in our experimentation is the relatively short context lengths used. We found that longer context lengths negatively affect the Uni[`MASK`] models' performance. In part, this could be addressed by designing masking schemes tailored to specific test-time tasks (see Appendix C), or using principled masking schemes [26]. However, this degradation may be attributed to our use of a BERT-like (rather than GPT-like) architecture, which seems less compatible with longer sequence lengths. A clear avenue of future work would therefore be to get the "best of both worlds": long sequences and benefits of `random-mask` pre-training by using a GPT-like architectures, with our `random-mask` and `finetune` training regimes. This requires finding ways to make GPT act like a bidirectional model. Recent methods in NLP might offer a useful starting point [1, 15], as has been explored by concurrent work to ours [27].

**Comparison to other applications of masked prediction and sequence models for sequential decision making.** In concurrent work, MaskDP [27] has also applied masked prediction to sequential decision-making. Similarly to Uni[`MASK`], MaskDP pre-trains a bidirectional transformer to predict randomly-masked token sequences corresponding to states and actions in a Markovian decision process. The main difference between our works is that we are more interested in comparing the performance between different training regimes, and testing the performance limits of having a single

---

[2]We additionally use input-stacking for Uni[`MASK`] – see Appendix D – but in preliminary experiments we found this to not affect performance.

set of weights to perform a large variety of tasks out of the box. MaskDP instead focuses on getting the best performance possible on a smaller subset of classic tasks (e.g. having separate architecture choices for RL). Future work could more systematically investigate the differences in our methods, e.g. how the MaskDP encoder-decoder architecture fares on multi-task performance as measured in our work. In addition, other architectural choices could be explored: in order to speed up training time efficiency, one could try to substitute BERT for XLNet or NADE-style approaches [44, 40]. Finally, another exciting direction for future work is determining whether the benefits obtained from `random-mask` (or even `multi-task`) apply to other types of inferences more generally (e.g. Bayes Networks); alternatively, even trivially extending the approach to multi-agent settings (for which token-stacking could prove more valuable), could enable interesting masking-enabled queries [29].

## 7   Conclusion

**Broader impacts.** The prospect of very large "foundation models" [2] becoming the norm for sequential problems (in addition to language) raises concerns, in that it de-democratizes development and usage [23]. We use significantly smaller models and computational power than similar works, leaving open the option to have more modestly-sized environment-specific foundation models. However, we acknowledge that this works still encourages this trend.

**Summary.** In this work we propose Uni[`MASK`], a framework for flexibly defining and training models which: **1)** are naturally able to represent any inference task and support multi-task training in sequential decision problems, **2)** match or surpass the performance of the corresponding single-task models after multi-task pre-training, and almost always surpasses them after fine-tuning.

## Acknowledgments and Disclosure of Funding

We'd like to thank Miltos Allamanis, Panagiotis Tigas, Kevin Lu, Scott Emmons, Cassidy Laidlaw, and the members of the Deep Reinforcement Learning for Games team (MSR Cambridge), the Center for Human-Compatible AI, and the InterAct Lab for helpful discussions at various stages of the project. We also thank anonymous reviewers for their helpful comments. This work was partially supported by Open Philanthropy and NSF CAREER.

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
