## A  Code

Our codebase can be found in [4]. Our code uses assets from gym-minigrid [8, Apache License 2.0], Decision Transformer [7, MIT License], and D4RL [16, Apache License 2.0].

## B  Training regimes

Each batch is made of many randomly sampled trajectory snippets. Across the tasks depicted in Figure 1, for each snippet $\tau_{t:t+k}$ we describe the input masking and predicted outputs in detail:

- **Behavioral Cloning.** Select $i \in [0, k]$ uniformly. Feed $s_{t:t+i}, a_{t:t+i-1}$ to the network (include no actions if $i = 0$), with all other tokens masked out. Have the network only predict the next missing action $a_i$.
- **Goal-Conditioned imitation.** Same as BC, but $s_{t+k}$ is always unmasked.
- **Reward-Conditioned imitation (Offline-RL).** Same as BC, but return-to-go $\hat{R}_t$ is always unmasked.
- **Waypoint-Conditioned imitation.** Same as BC, but a subset of intermediate states are always unmasked as waypoints or subgoals.
- **Future inference.** Same as BC, but the model is trained to predict all future states and actions, rather than only the next missing action.
- **Past inference.** Select $i \in [1, k]$ uniformly. Feed $s_{t+i:t+k}, a_{t+i:t+k}$ to the network, with all other tokens masked out. Have the network predict all previous states and actions $s_{t:t+i-1}, a_{t:t+i-1}$.
- **Forward dynamics.** Select $i \in [0, k-1]$ uniformly. Give the network the current state and action $s_{t+i}, a_{t+i}$, and have it predict the next state $s_{t+i+1}$. In theory, this could enable to handle also non-Markovian dynamics (we did not test this).
- **Inverse dynamics.** Select $i \in [1, k]$ uniformly. Give the network the current state and previous action $s_{t+i}, a_{t+i-1}$, and have it predict the previous state $s_{t+i-1}$.
- **All the above (ALL).** Randomly select one of the above masking schemes and apply it to the current sequence. This is a simple way of performing multi-task training.
- **Random masking (RND).** As we mention in section 3.1, for each trajectory snippet, first, a masking probability $p_{\text{mask}} \in [0, 1]$ is sampled uniformly at random; then each state and action token is masked with probability $p_{\text{mask}}$; lastly, the first RTG token is masked with probability $1/2$ and subsequent RTG tokens are masked always (see Appendix C for additional details). Randomly using the return-to-go in this fashion enables the model to perform both reward-conditioned and non-reward-conditioned tasks at inference time.

## C  The random masking scheme

Given the significance of `random-mask` in our work, let us take a closer look at the choices made in constructing this masking scheme.

A straightforward randomized masking would be to simply mask each of the state and action tokens with some fixed probability (in other words, fixing $p_{\text{mask}} = p$ for some constant $p$ rather than sampling it from $[0, 1]$). Indeed, this is a common masking scheme in NLP uses of BERT. However, in this scheme, the *number of masked tokens* is distributed as Binomial$(k, p)$ where $k$ is the context length. Then, the probability of almost fully-masked (or fully-unmasking) a trajectory snipped is exponentially small in $k$. This is an issue for us, since there are many meaningful tasks that require most tokens to be masked (past prediction) or unmasked (behavior cloning).

Our alternative distribution resolves this problem. In this distribution (described in the first paragraph of this subsection), the *number of masked tokens* is uniform in $[0, k]$.[3] In particular the tails are not exponentially small in $k$. Empirically, we found that this distribution works much better than the straightforward distribution described in the previous paragraph.

---

[3]See, for example, https://math.stackexchange.com/q/282347.

# D   Model architecture

**Input stacking.** An important hyperparameter for transformer models is what dimension to use self-attention over. Previous work applies it across states, actions, and rewards as separate tokens (or even individual state and action dimensions) [7, 21]; this can increase the effective sequence length that we would need to input a trajectory snippet of length $k$: for example if treating states, actions, and rewards separately (have self-attention act on each independently), the sequence length would be $3k$. While this is not an issue for the short context windows we use in our experiments, this seems wasteful: the main bottleneck for transformer models is usually the computational cost of self-attention, which scales quadratically in the sequence length.

To obviate this problem, we stack states, actions, and rewards for each timestep, treating them as single inputs. This way, we are making self-attention happen only across timesteps, reducing the self-attention sequence length required to $k$. This also seems like a potentially advantageous inductive bias for improving performance. Though we did not test this systematically, preliminary experiments did show that input stacking sometimes reduced validation loss.

**Return-to-go conditioning.**

Unlike previous work that considers return-to-go conditioning [7], we do not provide the model with many return-to-go tokens (one for each timestep). Providing just the first token should be sufficient for the model to interpret the return-to-go request (as, if necessary, the model can compute the remaining return to go in later steps). We found that this reduced overfitting for both `single-task` reward-conditioned training, or for `random-mask` training.

**Positional/timestep encoding.**

When conditioning on return-to-go tokens (i.e. for reward-conditioning), it is fundamental for the model to have information about what the time-horizon the specified return-to-go should be achieved by. To provide the model with this information [7] uses a "timestep encoding" instead of the standard positional encoding used in transformers: this consists of adding information to each input token which allows the model to identify to which trajectory-timestep such tokens correspond to.

One large downside of this is that adding timestep information directly in this manner greatly increases the tendency of the model to overfit. To obviate this problem, we use positional encoding (which only provides the model with information about the relative position of each token within each trajectory snippet $\tau_{t:t+k}$). However, making the change to positional-encoding in isolation would remove trajectory-level timestep information from the return-to-go token (the problem that "timestep encoding" was introduced to solve). To address this, we change the form of the return-to-go token to a tuple containing return-to-go and the current timestep, and find this to work well in practice.

# E   Minigrid experiments

Below we delineate some more details about our custom DoorKey Minigrid environment.

**Training dataset.**

We train Uni[`MASK`] models on training trajectories of sequence length $T = 10$ from a noisy-rational agent [46] which takes the optimal action most of the time, but has some chance of making mistakes proportional to their sub-optimality. More specifically, the agent takes the optimal action with probability $a \sim p(a) \propto \exp(C(a))$ where $C(a) = 1$ if the distance to the current goal (key or final goal) decreases, $-1$ if it increases, and $0$ otherwise.

**Environment.**

The state and action spaces are both represented as discrete inputs: there are 4 actions, corresponding to the 4 possible movement directions (up, right, down, left); taking each action will move the agent in the corresponding direction unless 1) the agent is facing a wall, 2) the agent is facing the locked door without a key. Stepping on the key location tile picks up the key. The state is represented as two one-hot encoded position vectors—the agent position and the key position (which is equivalent to the agent position once the key has been picked up). Both agent and key position have 16 possible values, some of which are never seen in the data (e.g. the agent position coinciding with a wall location). Together, such vectors are sufficient to have full observability for the task—as seeing the key location

coincide with the agent location informs the model that the agent is holding the key, and if the agent is holding the key it can open the locked door. Once the agent is holding the key, whether the door is open or closed is irrelevant.

Having the states and actions be discrete enables all model predictions to be done on a discrete space—which is particularly convenient as it enables the trained models to output any distribution over predicted states and actions, which can be easily visualized such as in Figure 6.

As the DoorKey environment have discrete actions and states, we use the softmax-cross-entropy loss over all predictions.

**Fixing prediction inconsistencies.**

In backwards inference, we note that sometimes the predicted state at the previous timestep may not be consistent with the dynamics of the environment or the observed states. In cases where the prediction is inconsistent with environment dynamics, we re-sample the prediction (rejection sampling). In cases where the prediction is inconsistent with the observed variables, we simply return the trajectory even though it may not be consistent with the conditioned states, although rejection sampling could also have been performed here.

**Hyperparameters.**

For each model and task, hyperparameters were obtained with a random-search method, which swept over batch sizes $(50, 100)$, token embedding dimensions $(32, 64, \text{or } 128)$, number of layers $(2, 3, \text{or } 4)$, number of heads $(4, 8, \text{or } 16)$, state loss re-scaling factors $(1, 0.5, \text{or } 0.1)$, dropout $(0 \text{ or } 0.1)$, and learning rates (selected log uniformly between $10^{-5}$ and $10^{-3}$). With number of layers, we refer to attention layers for transformers, and hidden layers for feedforward models. Optimal hyperparameter choice is reported in Table 2.

Each model was trained using the Torch implementation of the Adam optimizer. Training was performed over 6000 epochs with early stopping over the validation loss. Action:State loss indicates the relative re-scaling of the losses of actions and state predictions: we found it to sometimes be useful to offset the larger loss values of state predictions relative to action predictions (due to their larger dimensionality).

Each `finetune` model used the same hyperparameters as its corresponding `single-task` model, with the learning rate lowered to $5 \times 10^{-6}$ or $10^{-5}$, and the number of epochs to $500$–$6000$, depending on the task.

One thing to keep in mind is that while we search for hyperparameters which minimize the validation loss, this will not always correlate perfectly with reward, as showcased by prior work [20].

**Computational cost.**

Models were trained and evaluated on an on-premise server. The server has 256 AMD EPYC 7763 64-Core CPUs and 8 NVIDIA RTX A4000 GPUs. Running the experiments necessary to generate each of the heatmaps in the "Detailed heatmaps" section of Appendix F took approximately ten hours. We were rarely able to fully utilize the server (since it is shared with other projects), but we estimate that with full parallelization the models and data for each heatmap would take roughly half an hour to generate.

# F    Additional Minigrid experiments

### State-action distributions on MiniGrid

We visualize the distribution of states and actions for trajectories sampled from the model, conditioned on the initial state (essentially, looking at the transition frequencies of BC-sampled trajectories). As seen in Figure 8, the model learns to match the underlying distribution of trajectories of the agent (as can be verified by comparing to held-out data).

### Detailed heatmaps

We report below more validation-loss results from the Minigrid experiments. This section expands on Figure 7 by adding comparison to two baseline models: Decision Transformers (DT), and Uni[`MASK`] model implemented with a Multi-layer Perceptron architecture (trained with the same maskings as

| Model | Training task | Batch size | Embed dim. | Layer width | Num. layers | Num. heads | Action:State loss |
|---|---|---|---|---|---|---|---|
| Uni[MASK] single-task | Behavior Cloning | 250 | 32 | 128 | 2 | 4 | 1:0.1 |
| | Reward Conditioned | 50 | 32 | 128 | 3 | 4 | 1:1 |
| | Goal Conditioned | 250 | 128 | 128 | 3 | 8 | 1:1 |
| | Waypoint Conditioned | 250 | 128 | 128 | 3 | 8 | 1:1 |
| | Past Inference | 250 | 32 | 128 | 4 | 4 | 1:0.5 |
| | Future Inference | 250 | 128 | 32 | 2 | 4 | 1:0.5 |
| | Forwards Dynamics | 250 | 128 | 128 | 3 | 8 | 1:1 |
| | Inverse Dynamics | 50 | 128 | 128 | 3 | 8 | 1:0.5 |
| Uni[MASK] multi-task | (All the above) | 250 | 32 | 128 | 3 | 4 | 1:1 |
| Uni[MASK] random-mask | - | 100 | 128 | 128 | 2 | 8 | 1:1 |
| Decision Transformer | Behavior Cloning | 250 | 32 | 128 | 3 | 8 | 1:1 |
| Decision Transformer | Reward Conditioned | 250 | 32 | 128 | 3 | 8 | 1:1 |
| Multi-layer Perceptron | Behavior Cloning | 100 | 32 | 128 | 3 | - | 1:0.5 |
| Multi-layer Perceptron | Random Masking | 250 | 32 | 128 | 3 | - | 1:0.5 |

Table 2: Hyperparameters chosen for each model and training task. In addition to the column headers, the sweep found the best learning rate to be $10^{-4}$ and dropout factor to be $0.1$, in all settings. finetune models used the same hyperparameters as their corresponding single-task, and are therefore omitted.

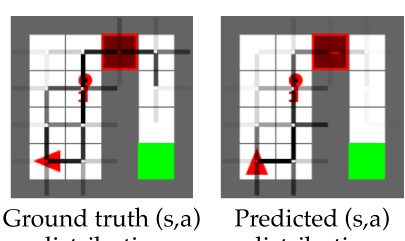

Ground truth (s,a) distribution   Predicted (s,a) distribution

Figure 8: Distribution of states and actions for trajectories in the validation set, vs. trajectories sampled from the model, conditioned on the initial agent position (1,4) and key position (2,2).

our transformer). We also vary the amount of data used to train each model (50, 1000, and the original 500 number of trajectories). We see that notwithstanding the differences in dataset size, the trends and relative orderings of performance between models tend to stay the same.

All results are reported across six random seeds. All standard deviations are on the order of or smaller than $0.01$, except for about four cells (in each data regime) with especially high mean losses. See Figures 9 to 14.

From these results, we see that using Decision Transformer in this context tends to slightly underperform relative to single-task Uni[MASK] models on the two tasks considered: Behavior Cloning and Reward-Conditioning. We also see that using an MLP architecture for one's Uni[MASK] model leads

to significantly worse performance than using our BERT-like transformer architecture: we suspect that this is due to the attention mechanism, which better lends itself to cleanly ignoring or using masked and unmasked information in the input.

**Decision Transformer with BC training.**

When reporting performance for Behavior Cloning using Decision Transformer (DT), we are training a DT model without inputting return-to-go information at training time. This ensures that the model should be trying to directly imitate the expert, rather than trying to achieve any specific reward. This is also the case for Appendix H.

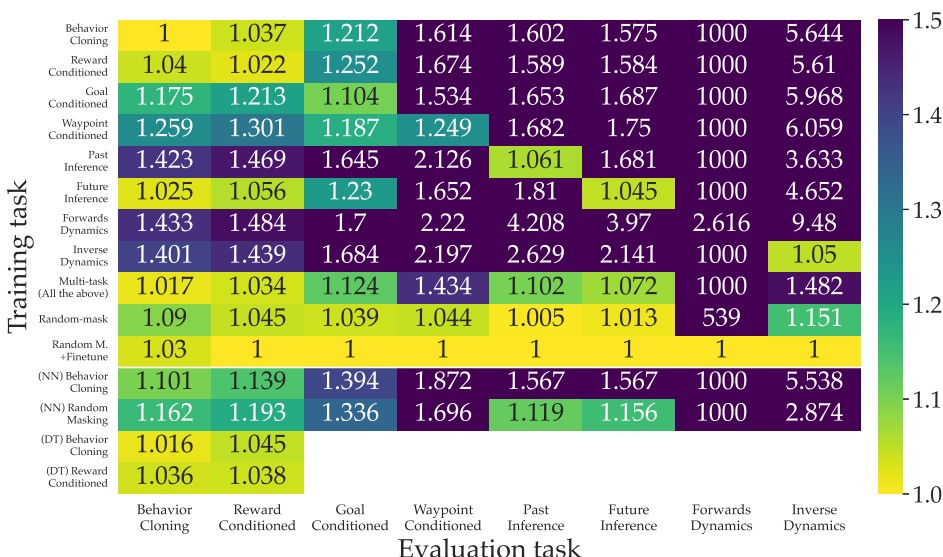

Figure 9: Same as Figure 7, adding the last four rows that compare to baseline models.

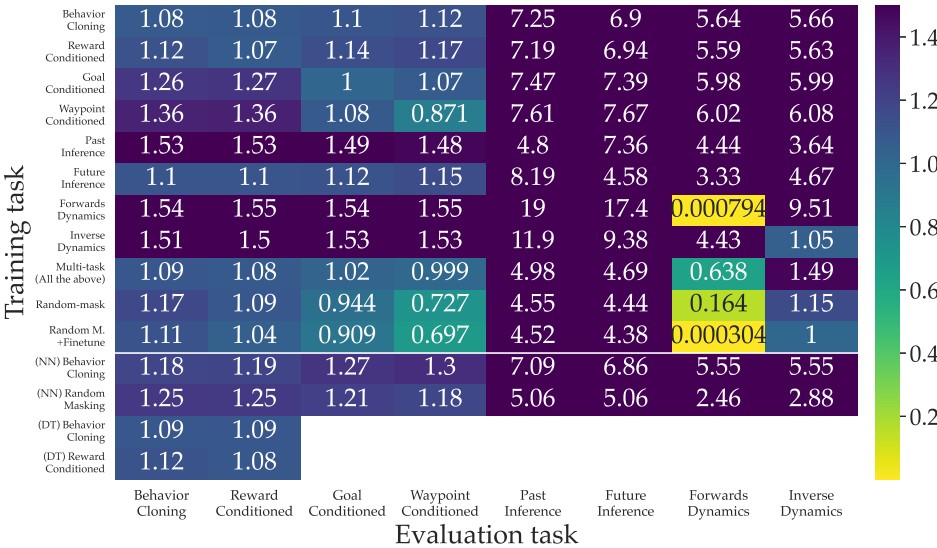

Figure 10: The raw loss values corresponding to Figure 9.

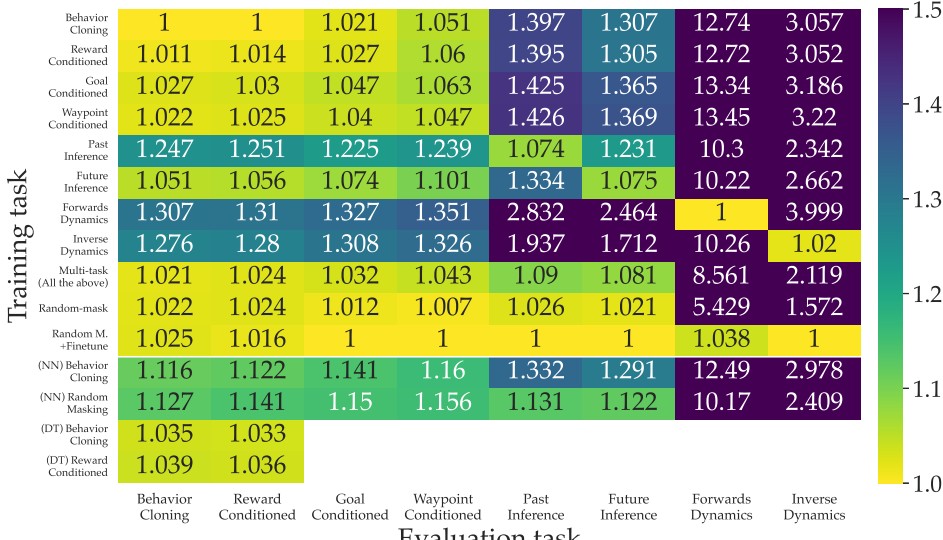

Figure 11: Figure 9 when using a dataset of 50 trajectories instead of 500.

| Training task | Behavior Cloning | Reward Conditioned | Goal Conditioned | Waypoint Conditioned | Past Inference | Future Inference | Forwards Dynamics | Inverse Dynamics |
|---|---|---|---|---|---|---|---|---|
| Behavior Cloning | 1.17 | 1.17 | 1.18 | 1.2 | 7.24 | 6.97 | 5.66 | 5.7 |
| Reward Conditioned | 1.19 | 1.19 | 1.19 | 1.21 | 7.24 | 6.96 | 5.65 | 5.69 |
| Goal Conditioned | 1.2 | 1.21 | 1.21 | 1.21 | 7.39 | 7.28 | 5.93 | 5.94 |
| Waypoint Conditioned | 1.2 | 1.2 | 1.2 | 1.19 | 7.4 | 7.3 | 5.97 | 6 |
| Past Inference | 1.46 | 1.47 | 1.42 | 1.41 | 5.57 | 6.57 | 4.58 | 4.37 |
| Future Inference | 1.23 | 1.24 | 1.24 | 1.25 | 6.92 | 5.73 | 4.54 | 4.96 |
| Forwards Dynamics | 1.53 | 1.54 | 1.53 | 1.54 | 14.7 | 13.1 | 0.444 | 7.46 |
| Inverse Dynamics | 1.5 | 1.5 | 1.51 | 1.51 | 10 | 9.13 | 4.56 | 1.9 |
| Multi-task (All the above) | 1.2 | 1.2 | 1.19 | 1.19 | 5.66 | 5.77 | 3.8 | 3.95 |
| Random-mask | 1.2 | 1.2 | 1.17 | 1.15 | 5.32 | 5.44 | 2.41 | 2.93 |
| Random M. +Finetune | 1.2 | 1.19 | 1.16 | 1.14 | 5.19 | 5.33 | 0.461 | 1.86 |
| (NN) Behavior Cloning | 1.31 | 1.32 | 1.32 | 1.32 | 6.91 | 6.89 | 5.55 | 5.55 |
| (NN) Random Masking | 1.32 | 1.34 | 1.33 | 1.32 | 5.86 | 5.98 | 4.52 | 4.49 |
| (DT) Behavior Cloning | 1.21 | 1.21 | | | | | | |
| (DT) Reward Conditioned | 1.22 | 1.21 | | | | | | |

Figure 12: The raw loss values corresponding to Figure 11.

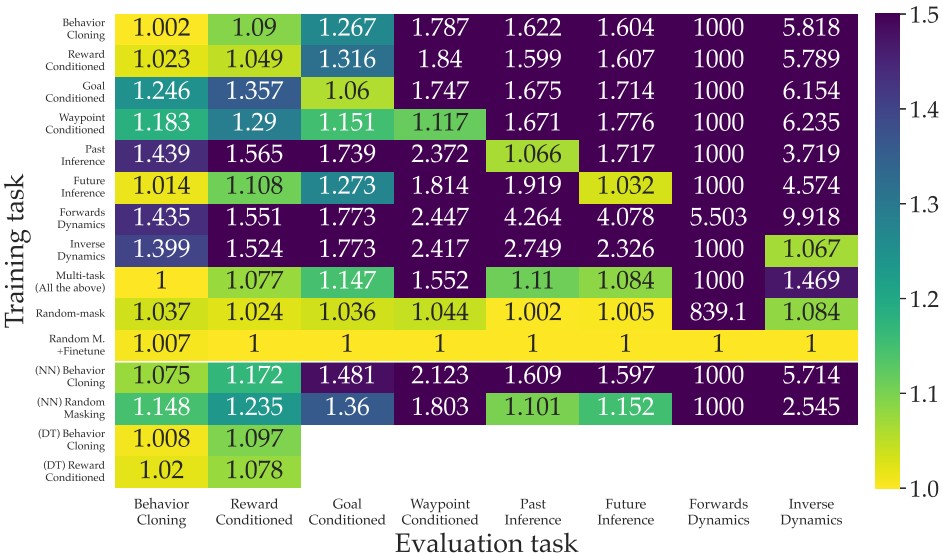

Figure 13: Figure 13 when using a dataset of 1000 trajectories instead of 500.

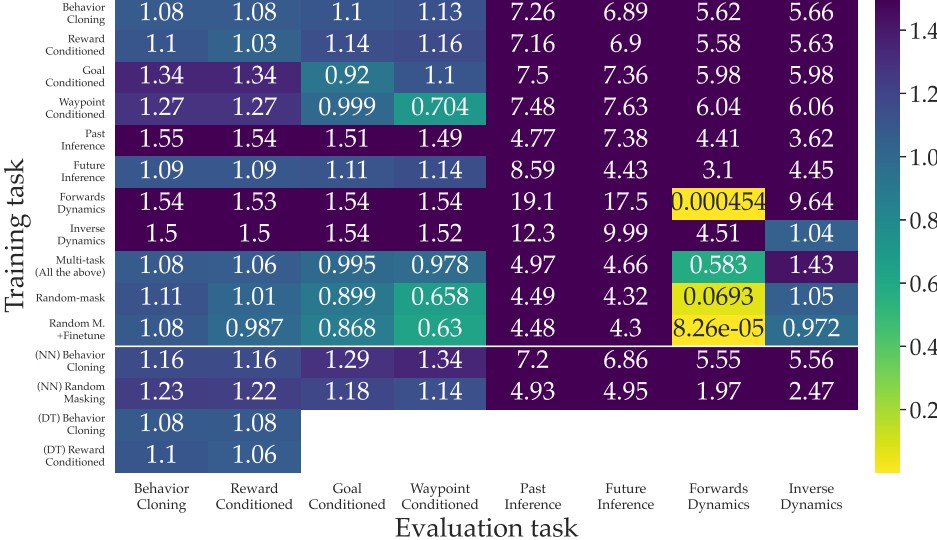

Figure 14: The raw loss values corresponding to Figure 13.

# G  Our Decision-GPT model

To obtain our Decision-GPT model, we use a standard GPT architecture (i.e. using a transformer decoder with *causal* self-attention), but incorporate the return-to-go and positional encoding design choices we used for Uni[MASK] models (which are described in Appendix D). This is to form an improved baseline from a simple GPT model.

# H  Maze2D experiments

**Choice of environment.**

We initially set out to compare the performance of Uni[MASK] on the same continuous control tasks used in [7]. However, after consulting with the authors of [7], we decided not to use the classic Mujoco control environments and associated D4RL datasets [16].

We list some of the issues with the D4RL datasets and classic control environments here: 1) given that the expert datasets were generated from Markovian policies and that these Mujoco environments are Markovian themselves, there is no direct reason for why sequence models should provide any benefit (although some benefit is observed in practice); 2) we noticed that completely overfitting a single trajectory with an MLP was sufficient for obtaining relatively good reward performance, indicating that such environments do not have enough inherent randomness to be a good indicator as to the generalization of trained policies – which is what we ultimately care about.

We tried to address these points in modifying the Maze2D environment. By choosing an environment in which the start and goal location are randomized, overfitting is not a viable strategy for generalization: memorizing a single trajectory in this setup leads to extremely poor performance.

As an additional detail, we modify the original environment reward to be dense, so that the reward at every timestep is given by the distance covered towards the goal.

See Figure 15 for example initializations of the environment.

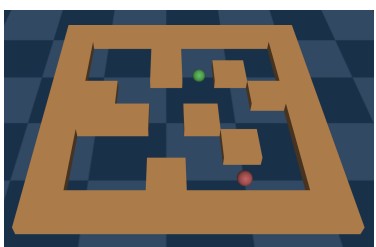 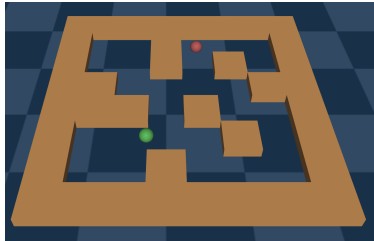

Figure 15: Examples of initializations in the Maze2D environment: the agent (in green) must navigate the environment to reach the goal (in red).

**Training.**

As the Maze2D environment has continuous actions and states, we use an L2 loss over all predictions. Each model was trained using the Torch implementation of the Adam optimizer. Training was performed for 2000 epochs with early stopping over the evaluation reward.

In reward-conditioned evaluation, choosing reasonable return-to-go (RTG) tokens on which to condition is non-trivial: asking for large reward in cases in which the goal is very close to the starting state leads to impossible-to-satisfy queries. Conversely, using the average reward as the goal return-to-go might be too conservative for easy initializations in which the point mass object can traverse most of the maze. To obviate this problem, we try to automatically determine what a reasonable RTG is at evaluation time using the following method: 1) reset the environment (leading to a random initial state); 2) find the trajectory in the dataset which has the most similar initial state (which also includes information about the goal location), and its total reward $R$; 3) Condition on an RTG of $1.1 \times R$. We found this behaves as intended qualitatively.

**Hyperparameters.**

For each model and task, hyperparameters were obtained with a random-search method, which swept over batch sizes $(50, 100, 200)$, token embedding dimensions $(64, 128)$, number of layers $(2, 3, \text{ or } 4)$, number of heads $(8, 16)$, state loss re-scaling factors $(1, 0.5, 0)$, and learning rates (selected log uniformly between $10^{-5}$ and $10^{-3}$). With number of layers, we refer to attention layers for transformers, and hidden layers for feedforward models.

Given that we found little difference between various hyperparameters across model types, the same set of hyperparameters was used across all conditions. The final hyperparameters are as follows: $10^{-4}$ learning rate, 100 batch size, 128 embedding dimension, 4 layers, 16 attention heads, and a state loss re-scaling factor of 1 (equivalent to no re-scaling).

Similarly to the Minigrid experiments, each `finetune` model used the same hyperparameters as its corresponding `single-task` model, with the learning rate lowered to $8 \times 10^{-5}$, and the number of epochs lowered to 600.

**Computational cost.**

We used the same compute infrastructure for our Maze2D experiments as in the Minigrid experiments (described in Appendix E). A training run in Maze2D takes approximately 4 hours on our server, but more than 20 runs can be run in parallel. In total, running all runs for Table 1 should take on the order of 10 hours when using our setup in parallel.