# OpenReview forum: "Uni[MASK]: Unified Inference in Sequential Decision Problems"
_NeurIPS.cc/2022/Conference — NeurIPS 2022 Accept_

### Official Review · Reviewer_SqCy · 2022-07-10

**Rating:** 9
**Confidence:** 5
**Soundness:** 4 excellent
**Presentation:** 4 excellent
**Contribution:** 4 excellent

**Summary:**

This submission proposed a new framework, Uni[MASK], which performs unified inference in sequential decision problems, using different masking schemes.

The authors demonstrated that randomly sampling masking schemes help the bidirectional transformer model being able to do behavior cloning, rewarding conditioning, dynamics modeling and more.

The experiments are first on grid-world and then extended to Mujoco-physics Maze2D environment. The comparison between single-task, multi-task, random-task, with finetune are through and solid, integrating the theory with practice, and quite convincing. The experiments in Mujoco-physics Maze2D environment shows that Uni[MASK] framework outperforms baselines, such as Feedforward Neural Network and Decision Transformer.

I fully endorse that this submission is a wonderful work: well designed experiments, diplomatic and sound illustration and the novelty of unifying decision making through Uni[MASK] and bidirectional models.

**Questions:**

From line 285 to line 2899, "A clear avenue of future work would therefore be to get the “best of both worlds”: long sequences and benefits of random-mask pre-training by using a GPT-like architectures, with our random-mask and finetune training regimes. This which would require finding ways to make GPT act like a bidirectional model – for which recent methods in NLP might offer a useful starting point". In the rebuttal period, may I know the opinion from the authors, why making GPT act like a bidirectional model is important in the decision making scenario? The answers to this question would make me (maybe other readers) understand this paper better.

**Limitations:**

* Relative short context lengths. Experiments with longer context lengths would make this submission more convincing.
* The experiments are mainly on the grid world (grid size is 4* 4). If the grid sizes increase more, this work would become more persuasive.

**Strengths And Weaknesses:**

Strengths:
* Uni[MASK] unifies inference task in sequential decision problems.
* The illustration of motivation, theory and experiments design is convincing and sound, written and presented very well.
* The authors demonstrate how randomly sampling masking schemes at training time produces a single multi-inference-task model that can do behavior cloning, reward-conditioning, dynamics modeling and more.
* The authors test how training on many tasks affects single-task performance.
* The authors show how fine-tuning models trained with random masking consistently outperforms single-task models.
* The proposed Hypothesis H1 ( {multi-task, random-mask, finetune} > single-task) and H2 ( {random-mask > multi-task} ) make sense intuitively, as "H1 test whether models indeed learn richer representations by training on multiple inference tasks" and H2 test "whether training on all possible tasks by randomly sampling maskings at training time is better than selecting a set of specific maskings"
* The results from well designed experiments strongly support hypothesis H1 and H2.
* The authors extensively analyze how different training regimes (combination of single-task, multi-task, random mask, fintune) affect performance. The analysis is very convincing to me.
* The authors also make experiments comparison with GPT-like architectures, and the analysis is also very convincing: "We find that while using GPT seems to yield similar performance to BERT for context length five, using GPT seems to give an advantage for longer sequence lengths," and "This suggests that if one were able to use a GPT architecture and train it with random masking and fine-tuning, it might be possible to get the best of both worlds."

Weakness:
* Relative short context lengths, and the experiments is mainly on the grid world (4 * 4), if the grid size increases, it might further improve this work.

---

> ### Author Response · Authors · 2022-08-02
> **Response to Reviewer SqCy**
>
> Thank you for your kind review! We hope that we can address your main concerns below:
>
> **Why making GPT act like a bidirectional model is important in the decision making scenario? The answers to this question would make me (maybe other readers) understand this paper better.**
>
> The task of predicting random missing tokens based on all inputted context requires bidirectionality, as the model should make use of future context for it’s predictions about missing tokens in the initial portions of the context. In our experiments, we show how random-mask+finetuning training consistently outperforms traditional single-task training (when using the same architecture for both training regimes).
>
> However, when comparing across architectures, this is not always the case. Consider Table 1 for context length 10: while random-mask pre-training still helps performance with the BERT-architecture, it is not sufficient to perform on par with the best GPT-architecture (which is not doing random-mask pre-training). We hypothesize that this may be because GPT models are better suited for generation tasks than out-of-the-box BERT models [4-5].
>
> In light of this, a natural next step would be to try to use random-mask training in combination with GPT-style architectures, in order to obtain the best of both worlds. However, given that standard GPT architectures are not bidirectional (as they use causal self-attention instead of simple self-attention), they are not compatible out-of-the-box with random-mask training. To make them compatible, one would have to use approaches such as the ones proposed recently  in [1-2].
>
> We will try to make the discussion around this point more clear in the final version of the manuscript.
>
> **Relative short context lengths**
>
> Regarding making context lengths longer, we would expect our current models to not perform very well because the number of possible maskings is exponential in the sequence length, so the benefits of richer representations (due to considering more tasks) would be outweighed by the sparsity of masking schemes which correspond to the test-time tasks we care about (which only increase linearly in sequence length). For handling longer sequence lengths, we expect better selected randomized masking schemes (potentially, by inventively readapting prior work [3] to our setting) and/or other model architectures (such as XLNet or bidirectional GPT-style approaches [1,2]) to be good first steps in addressing the issue.
>
> **Small minigrid size (grid size is 4 * 4)**
>
> This is a fair concern: we have increased the minigrid size in some extra experiments reported in the common author response.
>
> ---
>
> We hope the above was helpful to clarify some aspects of our work. We are happy to answer any other questions below!
>
> [1] Aghajanyan et. al. CM3: A causal masked multimodal model of the internet. CoRR, 2022.
> [2] Fried et. al. Incoder: A generative model for code infilling and synthesis. 2022.
> [3] Levine et. al., PMI-Masking: Principled masking of correlated spans, 2020
> [4] Alex Wang and Kyunghyun Cho. Bert has a mouth, and it must speak: Bert as a markov random field language model. 2019.
> [5] Elman Mansimov, Alex Wang, and Kyunghyun Cho. A generalized framework of sequence generation with application to undirected sequence models. 2019.

---

### Official Review · Reviewer_4afK · 2022-07-11

**Rating:** 7
**Confidence:** 3
**Soundness:** 3 good
**Presentation:** 3 good
**Contribution:** 3 good

**Summary:**

This paper presents the Uni[MASK] framework, which generalizes masked language modeling (in natural language processing) to more general sequence decision problems. Given actions, states and optional property tokens (such as the return-to-go), multiple tasks such as behavioral cloning, reward-conditional offline RL, forward dynamics, etc., may be represented by specific masking schemes. The authors propose to train a model by using a randomized masking scheme (instead of single-task only or with a fixed set of tasks). Such a Uni[Mask] model may also be fine-tuned on a given task afterwards. A pre-trained Uni[MASK] model often performs comparably to single-task training, and fine-tuning further improves results.

[Post author response] I updated the rating I assigned to this paper, in particular because of the correction to the originally misleading discussion.


**Questions:**

**Questions**

Most important question: Could you please clarify exactly what you mean by "BERT" architecture vs "GPT" architecture, and how they interact with the masking schemes you propose? In NLP, BERT uses a transformer encoder (self-attention) with some masked inputs whereas GPT uses a transformer decoder (causal self-attention, i.e. not looking at the future).

L160+: Should there be a reward for shorter paths? Otherwise, there could be useless

L190: Why no test data?

L212: Do you mean figure 15 in the appendix? In general, figures referred in the main text should also be within the paper itself.

**Suggestion**

Please move the description of random masking (end of appendix 1) and appendix 2 to the main text. This is too important to leave out.

**Other**

L235: This points to the wrong appendix.

**Limitations:**

Some limitations are discussed.

**Strengths And Weaknesses:**

**Strengths**

The proposed approach neatly formalizes multiple tasks under a general masking framework.

Training a Uni[MASK] model, then finetuning it on a given task, generally leads to better results than single-task training only.

The approach is tested in both discrete (MiniGrid) and continuous (Maze2D) environments.

**Weaknesses**

The discussion around line 200 is misleading. In figure 7, multi-task performs worse than single-task (although sometimes negligibly), which doesn't support H1.

The distinction between BERT-like and GPT-like models should be described more clearly (see question below).

The MiniGrid environment appears to be very simple. I would be interested in seeing how well the approach scales with larger grids.

---

> ### Author Response · Authors · 2022-08-02
> **Response to Reviewer 4afK – Part 2**
>
> **L190: Why no test data?**
>
> We acknowledge this could be an improvement to the evaluation process. Even though only using a validation set is the standard in the field [1-2], we agree that we should try to improve the field to uphold a better standard.
>
> In our experiments almost all our hyperparameter tuning was performed without looking at the validation loss. When we did look at it, it was across all conditions (our main method and baselines alike), so on a baseline there is no reason to expect that this was to our advantage. If the paper is accepted, we will confirm before camera-ready time that all our conclusions still hold on an independently sampled test dataset.
>
> **L212: Do you mean figure 15 in the appendix? In general, figures referred in the main text should also be within the paper itself.**
>
> Thanks for pointing this out! We had moved the figure from the main text to the appendix, and missed this figure reference. We have updated the reference to simply point to the relevant appendix section.
>
> **Please move the description of random masking (end of appendix 1) and Appendix 2 to the main text. This is too important to leave out.**
>
> We agree with you that these are important sections. Because of space limitations at rebuttal time we weren’t able to move everything to the main text. However, we tried to add further information in Section 3.2.1 about how the random masking is performed, and updated Appendix A.2. If the paper is accepted, we will use the extra page to incorporate the information still missing in the main text.
>
> **L235: This points to the wrong appendix.**
>
> Fixed in the updated version – thanks for catching this!
>
> ----
> We hope our comments helped clarify some aspects of our work. We are happy to have further discussions about any of the above points!
>
> [1] Hussenot et al. “Hyperparameter Selection for Imitation Learning.” ICML (2021).
> [2] Scott et al. “RvS: What is Essential for Offline RL via Supervised Learning?” ICLR (2022).

---

> > ### Comment · Reviewer_4afK · 2022-08-09
> > **Reply to authors**
> >
> > Thank you for your response. It has addressed most of my concerns. I personally consider Transformer encoders and decoders (without cross-attention) to be the same architecture, but with a different 2-D mask (used in a difference sense than input [MASK] tokens) of shape sentence_length x sentence_length that determines which representations are attended to.
> >
> > I will take some time in the next few days to go over the updated version and potentially update my assessment of the paper.

---

> ### Author Response · Authors · 2022-08-02
> **Response to Reviewer 4afK – Part 1**
>
> Thank you for taking the time to write your helpful review and detailing your concerns. We hope they are addressed in our comments below.
>
> **The discussion around line 200 is misleading.**
>
> Firstly, we’d like to thank you for finding the inconsistency in the discussion of results around line 200. We’ve fixed this in the latest version of the paper which we have uploaded. We rewrote the relevant part of Section 4.3, clarifying that “Overall, these results don't fully support H1, given `multi-task`’s poor performance and `random-task`'s performance which is not consistently better than `single-task`”.
>
> **Could you please clarify exactly what you mean by "BERT" architecture vs "GPT" architecture, and how they interact with the masking schemes you propose?**
>
> We use the terms BERT and GPT similarly to NLP, exactly like you said: when we say we use a BERT architecture, we use a transformer encoder (self-attention) in which we mask inputs (randomly at training time, and in structured ways at test time); whereas, when we say we use a GPT architecture, we use a transformer decoder (with causal self-attention, i.e. not looking at the future).
>
> More specifically, to obtain our Decision-GPT model, we use a standard GPT architecture (i.e. using a transformer decoder with causal self-attention), but incorporate the return-to-go and positional encoding design choices we used for Uni[Mask] models (which are described in Appendix A.3). We have updated Appendix A.6 to be more clear about this.
>
> **The MiniGrid environment appears to be very simple.**
>
> We acknowledge that the minigrid environment we consider is quite simple. We chose it because it is meaningful to evaluate all the various tasks that we were considering on it. It was also a discrete state and discrete action environment, enabling our model to perfectly capture output uncertainty by representing categorical distributions over states and actions.
>
> The simplicity of the MiniGrid environment is complemented by the more complex Maze2D environment, as it is partially-observable (velocity is not directly observed) and has a continuous state and action space. That being said, we agree that training on more complex environments is an exciting future application.
>
> **I would be interested in seeing how well the approach scales with larger grids.**
>
> Thanks for your suggestion! We tried experimenting with a larger grid and reported results in the general response to all reviewers.
>
> **L160+: Should there be a reward for shorter paths?**
>
> For our purposes we didn’t find a reward for shorter paths necessary: if one wants optimal behavior, one can request the maximum amount of reward, together with the agent being at the goal at the first possible timestep. While this is non-standard, in the minigrid we were mostly concerned with showing how flexibly one can specify desired behaviors to our model.
>
> Although we haven’t tested this, one could make there be a higher “reward for shorter paths” by applying a discount factor ($\gamma<1$) to the environment rewards before calculating the RTGs used at training time. That way, two paths that would have the same reward under $\gamma=1$ (one shorter, one longer) would have different discounted rewards, making it possible to request the shorter path from the model using RTG alone.

---

### Official Review · Reviewer_YeV6 · 2022-07-11

**Rating:** 8
**Confidence:** 4
**Soundness:** 3 good
**Presentation:** 4 excellent
**Contribution:** 3 good

**Summary:**

Proposes an approach to using a single bidirectional transformer to accomplish many different reinforcement learning tasks, including behavior cloning, (sub)goal conditioning, and conditioning on properties more generally. It accomplishes this by creating a sequence of out of the (state, action, properties) for all timesteps, then either (1) training on a single type of mask, (2) training across many masking styles drawn from fixed set, and (3) randomly masking positions in the sequence. (2) and (3) can be finetuned in single task fashion. They evaluate on a toy gridworld and the mujoco 2d maze tasks. They find that Uni[MASK] performs better across both given a short truncated sequence, but it underperforms a GPT model for longer sequences. They attribute this to difficulties in training BERT models.

**Questions:**


	1. Clarify which task is shown in table 1. The grid world or maze?
	2. Consider expanding on what position vs. timestep encoding is in the body of the paper (line 231)?
	3. The "bert has no mouth and must speak" paper was withdrawn
	4. Take a look at XLNet and NADE style masking, which would allow combining benefits of GPT and BERT
	5. One possibile reason the longer sequence is failing is because it is never exposed to many masked positions in a row at the end of the sequence. Randomized decoding as in xlnet might help here, or applying more structured masking (such as blocking out spans instead of ). See the uni
		1. Perhaps training on both multitask and
	6. What happens if you train on both multitask and random masking?
	7. Potential improvements
		1. Vary the sequence lengths instead of holding fixed
			1. This is more natural for GPT style models, but can be done with XLNet/NADE approaches as well
			2. How might Uni[MASK] handle the full length 200 timestep sequences instead of truncating to last 5/10?
		2. Ideally would have more complicated tasks than the two presented
		3. Consider studying how
	8. Relevant concurrent work: foundation posterior, which applies a similar approach to inference in probabilistic programs (i.e. generalized bayesian networks) in Stan.
	9. nits
		1. works -> work in line 300

**Limitations:**

There are no major concerns with this paper

**Strengths And Weaknesses:**


	- originality
		- Concurrent work suggests that the core ideas of this paper are in the air (see: foundation posterior, multigame decision transformer, The UL2 paper on unifying masking schemes).
	- quality
		- The paper is well executed. It has simple experiments which verify the core claims.
		- My main concern is that the length 10 context experiments underperforming. Would like to see more study of this instead of alluding to claims that BERT has this known failure mode.
	- clarity
		- The paper is well written and a pleasant read. It adequately cites prior work, and it does a great job of explaining and illustrating the core ideas.
	- significance
		- The paper is a solid contribution to the literature. Overall, its core findings and technique align with concurrent and prior work.

---

> ### Author Response · Authors · 2022-08-02
> **Response to Reviewer YeV6 – Part 1**
>
> Thank you for engaging with our work and for your detailed review!
>
> We attempt to address your questions below:
>
> **1. Clarify which task is shown in table 1. The grid world or maze?**
>
> Thanks for catching this – it was unclear from the table caption: we have updated the manuscript to more clearly specify that Table 1 refers to the Maze2D results.
>
> **2. Consider expanding on what position vs. timestep encoding is in the body of the paper (line 231)?**
>
> We agree that this is important context: we have expanded our mention of this in Section 3.2, with all other relevant details being provided in Appendix A.3.
>
> **3. The "bert has no mouth and must speak" paper was withdrawn**
>
> Thank you for pointing this out! According to [their errata](https://sites.google.com/site/deepernn/home/blog/amistakeinwangchoberthasamouthanditmustspeakbertasamarkovrandomfieldlanguagemodel), the paper was retracted because of an error in their proof that BERT can be thought of as a Markov Random Field. In that sense, their empirical experiments on using BERT for language generation should still be valid (as they don’t depend on the proof in any way). Another work that builds off of the fact that BERT will not be good for language generation without modification is [1] – we have also included that in the new version of the paper. That being said, they also cite “BERT has a mouth, and it must speak”.
>
> **4. Take a look at XLNet and NADE style masking, which would allow combining benefits of GPT and BERT**
>
> We became aware of XLNet/NADE style masking during this work and have considered attempting to use that for future versions of the model. Unfortunately, we only thought of it after we had committed most of our infrastructure to using the BERT model. We have updated our paper’s future work section to include this direction.
>
> **5. One possible reason the longer sequence is failing is because it is never exposed to many masked positions in a row at the end of the sequence. Randomized decoding as in XLNet might help here, or applying more structured masking.**
>
> We agree that potentially using XLNet-style masking could help with this problem.
>
> Regarding more structured maskings, this is actually how we started out performing experiments: we investigated blocking out structured spans of tokens quite extensively but found that the model didn't perform significantly different from the other masking schemes (multi-task, etc.). Specifically, we were running experiments masking random contiguous subspans of states and actions (under various configurations).

---

> ### Author Response · Authors · 2022-08-02
> **Response to Reviewer YeV6 – Part 2**
>
> **6. What happens if you train on both multitask and random masking?**
>
> We did preliminary experiments on this but didn’t see any significant differences in performance. Given that the training regime was more complex but didn’t have visible advantages, we shifted our focus away from that. Note that our preliminary investigations were carried out in the MiniGrid environment - it is possible that such a mixed regime is advantageous in different settings.
>
> **7. Potential improvements: Vary the sequence lengths instead of holding fixed:**
>
> We agree that all the avenues for potential improvements suggested by you are very worthwhile future investigations.
>
> Firstly, just in case this is a misunderstanding, we’d like to clarify that the results in Table 1 are with different sequence lengths (“Ctx len”), of 5 and 10. However, we realize these are quite short
>
> Using XLNet / NADE would be a very interesting direction, but we expect that porting these implementations to the sequential decision-making domain to be a sufficient amount of work to be its own paper.
>
> In preliminary experiments in which we increased the sequence length to be >20, we saw decreases in performance, which are likely due to the random-masking scheme being too extreme in that setting: since the number of possible maskings is exponential in the sequence length, then the benefits of richer representations (due to considering more tasks) seem to be outweighed by the sparsity of masking schemes which correspond to the test-time tasks we care about (which only increase linearly in sequence length). For longer sequence lengths, we expect better selected randomized masking schemes [2] or other models (as you suggested) to be good first steps in addressing the issue.
>
> **7. Potential improvements: Ideally would have more complicated tasks than the two presented**
>
> We agree that this is a limitation of our work. While we tried to cover many desirable properties (continuous/discrete states and actions, fully/partially observable environments) and simultaneously make our experimentation loop fast, we acknowledge that it would be interesting to scale this approach further to truly test its limits. We hope that the additional experiment we ran on a larger version of minigrid, although not necessarily "more complicated" (the environment dynamics are the same), might help address your concerns.
>
> **8. Relevant concurrent work**
>
> Thank you so much for pointing us to relevant concurrent works! We’ve updated our manuscript to include them.
>
> Feel free to ask us for any more clarifications!
>
> [1] Mansimov et al. “A Generalized Framework of Sequence Generation with Application to Undirected Sequence Models.” 2019
> [2] Levine et. al., “PMI-Masking: Principled masking of correlated spans”, 2020

---

### Author Response · Authors · 2022-08-02
**General Response to Reviewers**

We thank the reviewers for their insightful and constructive feedback!

We are encouraged that you had an overall positive impression of our work: saying that it “neatly formalizes multiple tasks under a general masking framework” (Reviewer 4afK) and constitutes a “solid contribution to the literature” (Reviewer YeV6). We also appreciate that Reviewer SqCy thought this paper was a “wonderful work” with “well designed experiments, diplomatic and sound illustration and the novelty of unifying decision making through Uni[MASK] and bidirectional models”.

**Missing information from the main text**

Some of your concerns relate to missing information from the main text which is important for the understanding of the paper: we have tried our best to incorporate some of this in the newly uploaded version, but haven’t been able to do so fully because of space constraints. That being said, we’d like to emphasize that all relevant details needed to reproduce the work are still included in the Appendix and the source code will be made available by the time of the conference if the paper is accepted. If the paper is accepted we will make sure to incorporate everything you requested into the main manuscript (as the camera ready version can have an extra page of content).

**Bigger grid for the minigrid task**

In terms of the experimental setup, Reviewer YeV6 mentioned that more complex tasks could have further strengthened our work. Reviewers SqCy and 4afK specifically point to the small size of our minigrid as one area of possible improvement. In light of this, we ran one additional set of experiments with a larger minigrid (equivalent to Figure 9 in the Appendix, but for a larger minigrid).

The bigger minigrid is 16 by 16 instead of 6 by 6. Utilizable state space (not occupied by walls) is 13 for prior experiments (4 by 4 minus interior walls) vs 183 for the current ones (14 by 14 minor interior walls), meaning this is a ~14x increase in state space. The mechanics of the environment are the same (there is a key, a locked door, and a goal location). Context length was left at 10 timesteps, as changing it would have required hyperparameter tuning (which would have been tough given time constraints of the rebuttal period). Training hyperparameters were left unchanged with the exception of making some runs longer.

The training data was collected in a similar approach to the one in the smaller minigrid. The only difference was that we increased the rationality parameter of the expert to be slightly higher before running any experiment (we did not tune the parameter), so that the expert would be slightly less random and make more progress in the task within the 10-timestep horizon.

**Results figure [here](https://i.postimg.cc/1RLs4f5Q/myplot.png)**. Let us know if you have trouble accessing it – we couldn't include it directly in the response.

We see that the results mostly follow the same trends as those in the main paper, with the main difference being that multi-task training performs somewhat better on behavior cloning, reward conditioning, and goal-conditioning (at the expense of worse performance on past and future inference). This strengthens the support of H1 relative to Figure 9, while still supporting H2 for the most part (in 6/8 tasks, random mask training performs better than multi-task).

We hope you find these additional experiments a sign of the applicability of our method.

------

We try to address your individual concerns below, and are happy to answer any further questions.

---

### Meta-Review · Area_Chair_fmhP · 2022-08-20

**Recommendation:** Accept
**Confidence:** Certain

**Metareview:**

This paper extends masked language modeling to other sequential decision making problems. The idea is simple (which is a plus), the experiments are thorough, and the results are convincing. All reviewers agreed this is a good paper. I recommend acceptance.

**Award:**

No

---

### Decision · Program_Chairs · 2022-09-14

Accept